# A Narrative Exploration of One Teacher's Storied Experiences of Online Learning during COVID-19

Joanna Mei Lin Lim

Independent Researcher, Christchurch 8140, New Zealand; joannameilinlim@gmail.com

**Abstract:** The hubbub of teaching lives is enriched by the relationships between students, colleagues, parents and the larger schooling community. When these relationships are disharmonious, attending to the dissonance within these relationships may offer insight into teachers' relational work. This autoethnographic article focuses on one international elementary school teacher's experience of teaching online in West Africa during COVID-19. Teaching online in a developing country with political and physical instabilities compounded the chaotic experience of living and working during the pandemic. Guided by this research question, "How did online learning impact my relationship with students?", the author utilized writing as an inquiry approach to make sense of the challenging aspects of her online teaching experiences. By delving into narrated moments, the author engaged in reflexive analysis of storied experiences. This process illuminated the meaning-making steps that she took to appreciate the nuances contained within specific moments that she had with a student and his father. Through storying and re-storying these moments, the author wondered and metaphorically wandered to unearth potential emotions, assumptions and motivations that permeated her experiences. Such an explorative focus on teachers' subjective meaning-making process augments the wider body of work on online education and, in particular, this study's inquiry into the complexity of educational relationships through a narrative lens offers insights into the inner workings of teacher emotions and feelings. This paper reveals how teachers may benefit from adopting a reflective and reflexive sense-making approach towards understanding their emotions, feelings, responsibilities and relationships with students and parents, especially during a time of crisis. This article contributes to the ongoing discussions about the complexities of teachers' relational work and it enriches the extant literature on online education by shedding light on the individualistic ways that teachers cope with the uncertainties of teaching during a time of crisis. Deeping our collective understanding of how teachers cope can help us to provide better support for teachers and students during crises such as COVID-19.

**Keywords:** narrative inquiry; teacher narratives; online learning; teacher and student relationships; COVID-19; emotional and relational work

## 1. Introduction

Teaching experiences are influenced by the connections that teachers make with their learners. Understanding the subtleties within these connections is critical because they may include affirmative or adverse feelings and emotions that shape teachers' experiences [1]. One avenue of investigation into teaching experiences could be through teacher stories, which enables teachers to explicate their sense-making process. Delving into the inner workings of teaching lives illuminates how teachers make meaning of their experiences; these meanings may make visible insightful ways for us to better understand and support teachers and their well-being.

This autoethnographic paper is about one elementary school teacher's inquiry into her online teaching experiences during COVID-19. Due to mandatory quarantines, travel bans and social distancing protocols during the 2020 school year, the author started teaching online in mid-August 2020 and taught sixteen third-grade (eight and nine years old)

students. As was commonplace with international schools, these sixteen students belonged to eleven different nationalities even though most of them were physically located in the same country. This school was located in West Africa and catered to Pre-Kindergarten (two year olds) to Grade 12 (eighteen year olds) students. It was a small (approximately 130 students), private, community run, not-for-profit school that provided an American-style education to foreign and local nationals. This school was located in a politically unstable country that had frequent power, water and internet disruptions.

These contextually bound factors placed unique organizational, cultural and physical challenges that affected the author's experience in addition to the chaos caused by COVID-19. They also inspired the author to draw upon Richardson's (2000) [2] writing as an inquiry approach to document and analyse her experiences of teaching online. With the research question, "How did online learning impact my relationship with students?", the author sought to detail the reflective and reflexive methods that she used to negotiate the challenges of developing educational relationships during online teaching. In this paper, the author narrowed her narrative lens to analyse the relational dynamics between her, a student and his father.

## 2. Teacher Stories as a Way of Making Sense of Lived Experiences

Storytelling as a humanistic endeavour has brought attention to the ways that we communicate, connect and, on an emotional plane, commiserate. The narrative turn [3] indicated the ways in which stories become narratives or research data, and the ensuing use of narrative methods of analysis that are purposefully used to elicit meaning. Turning stories into narratives can be a confronting, idiosyncratic and emotion-laden process. When narratives are used to expose the inner workings of teaching lives, it allows teachers to inquire, as an insider, who tells the story, whilst adopting the role of a researcher and an outsider, who meticulously unravels the threads of meaning within the experience. It is this autoethnographic space that Richardson (2000) [2] celebrated as a valuable space for inquiry. Following in her narrative writing method of inquiry, teachers can be encouraged to inquire into their experiences by writing to reflect and then analysing reflexively to explore how they managed themselves within the experience. Whilst inspired by Richardson's (2000) [2] autoethnographic writing as a space for inquiry, this study departs from autoethnography in the way that it focuses on the author's sense-making explorations. These narrative explorations borrow the qualities of autoethnography (a study of the self in relation to a broader socio-cultural/socio-political context) and self-study (a study of the self that is framed within an educational context) in the way that they explore the self [4], but, ultimately, this study is anchored to its narrative intentions and roots because it is a narrated exploration of lived, storied experiences of one teacher.

When teachers tell their stories, they open their lives up to examination. Paying attention to storied experiences encourages teacher voices to be recognised, valued and prioritized as a site of knowing [5]. When teacher stories are restructured into artefacts for examination, they are formalised into narratives [6] which can be viewed as snapshots of insider perspectives of the educational experience. This paper builds upon an existing body of work into teacher stories and narratives [7–9] and it delves into the inner sanctum of teaching lives. Its particular focus on online learning experiences magnifies the complexities of teaching relationships, thereby enriching existing work on understanding online student relationships and engagement [10–13] through a narrative lens.

Ongoing narrative work on teaching lives has produced a diverse body of work. Barkhuizen and Consoli (2021) [14] aptly called for teacher narratives to be contextualised and pursued with a close eye on the sense-making journeys that teachers take. While the fundamental elements in narratives can be whittled down to their settings, plot and characters, acknowledging teachers as storykeepers and storytellers allows them to intentionally tell stories to relay meaning, embolden practical ways of knowing and engage others in reflective musings about relational work. Any effort to investigate the relational perspectives that flush out the "response-ability" [15] (p. 249) that teachers need in order

to nurture meaningful learning is helpful because teacher narratives bring to the fore the relationships that teachers can have with people, places or objects [7]. Capturing these narratives celebrates the emotional, relational and psychological labour that connects teachers to their communities, and it also permits contextualised insights into authentic teaching and learning ecosystems. When Clandinin and Connelly (1998) [16] coined the term "stories to live by" to capture the landscape of school reforms, they paved the way for storied teaching realities to be recognised as a legitimate way of studying the complexities of schooling. Since then, the body of work around teacher narratives has blossomed and touched upon a multitude of layers within teaching lives [5], providing an opportunity to study the narrated truths within individual experiences [17].

Stories that become research narratives are the objects under scrutiny. However, stories that we live by, as Caine et al. (2013) [5] described, are framed within the lived and re-lived experiences. Thus, to inquire narratively is akin to seeking the humanistic essence that lives within words, transcending the physical, temporal and social space. Storying experiences into narratives turns these storied moments into research artefacts that pause time by allowing memories to be articulated and reconnected to assumptions and presumptions. While dissecting memories can be cathartic, which can be imagined as a mental cleansing or purging process, likewise, the process of studying narratives can be mentally and emotionally draining, because it demands the researcher to self-interrogate the assumptions, beliefs and values that underpin and permeate thoughts and actions.

From a mental health and recovery narrative perspective, storytelling and sharing have gained recognition as remedies that reject the hard, rational and cold distance purported within the science of medicine. In their systematic review of mental health recovery narratives, Llewellyn-Beardsley et al. (2019) [18] found that recovery narratives were fluidly constructed to communicate meaning to others. While this paper is by no means a recovery narrative, it is inspired by the notion that teachers can write, reflect and muse in order to make sense of challenging experiences. In this paper, I examined how teachers can use storytelling to better understand themselves and their practice, with an additional focus on storytelling as a form of healing or a means to manage difficult experiences.

In their investigation into the complexity of teacher and student relationships through narratives, Rytivaara and Frelin (2017) [19] underscored the commitment that teachers put into connecting and building trusting relationships with others. In my case, I needed to better understand my own commitment to educational relationships in order to understand my experiences. By retelling my story, I imagined, I re-lived and I paid close attention to my unexamined sense-making process in order to unpack the chaotic and stressful moments that I had when I was teaching online during COVID-19. Based on this study, I believe that teachers could benefit from learning how to tell and write personal stories in order to cope with the demands of relational work that is embedded in teaching.

## 3. The Methodological Process

To preserve the confidentiality of the relationships that will be discussed, pseudonyms have been used throughout the study. Relevant details about the school, the student and his father will be provided to give readers a contextual and background understanding of this study. It is relevant to state that the primary aim of this study is to interrogate the singular, narrated perspective of the author in order to illuminate her meaning-making process.

As a teacher–researcher, I wrote and analysed my lived experiences over a period of two years, from 2020 to 2022. I took mindful steps to protect the identity of my student and his father by focusing on my interpretation of events. The research question that guided the study, "How did online learning impact my relationship with students?", was aimed at investigating how teachers such as myself had coped with the effects of pandemic-compelled online education. Commencing with pieces of journal reflections that grew into longer fragments that were stitched together, these moments morphed into paragraphs of thoughts that were dissected, rested and reframed as lines of inquiry into the stories that I told about my online teaching experiences. This process of storying and re-storying my

experiences encouraged me to identify the ways in which my storied voice differed and it challenged me to question aspects of myself from my motivations, reactions, beliefs to values. The resulting sense-making process became a repetitive cycle of writing to question as well as writing to clarify the meaning that I had attached to events.

Within this explorative dance of thinking, feeling and naming my experiences, the narrative methodology that I subscribed to fell within the methodological realm of narrative inquiry, which I defined as inquiring ontologically (viewing the world narratively) and epistemologically (framing narrated experiences as a way of knowing experiences) into experiences [5]. In my reflective space, I revisited my memories through raw moments as well as unrealized aspects of the stories I carried with me. Through a fluid, interrogative lens that was inwards, outwards and sideways, I looked for the cracks, the obvious and the hidden. Only by pushing beyond the obvious was I able to uncover, consider and question the nuanced meanings that I had attached to my experiences.

This narrative study applies aspects of autoethnographic, self-study and narrative approaches [4] to examine the experience of one teacher; in this case, the researcher is the teacher under study. It is the "truth of educators' unique but similar experiences" that studies such as this can offer to the body of knowledge about teaching lives [20]. Going beyond the idea of ego-centric, navel-gazing conceptions, an individualistic focus celebrates the depth and nuanced analysis of the inner sanctums of teaching minds. To uphold measures of rigor and trustworthiness, the author focused on narrative verisimilitude, working towards the notion that the moments highlighted would resonate with other teachers; verisimilitude (the quality of believability) and resonance are criteria well suited for alternative, qualitative methodologies that privilege idiosyncratic meanings [21]. What will be presented in this paper are insights, musings and probabilities that invite the reader to question and engage in discussions about the possibilities that percolate in teachers' minds when they attempt to make sense of their experiences.

For the sake of clarity, I italicised the voice of my remembered and analysed self, because this was the reflective voice that told and retold my stories. Through re-storying, I attempted to capture, analyse and reimagine my memories in order to uncover the meaning that I had attached to those memories. This repeated process of getting to know myself through memories encouraged me to seek out the significance that I had attached to the relationship that I had with Oscar and his father. By delving into my relationship with Oscar, I gradually realized how my inner turmoil had coloured the stories that I carried about Oscar and his father. The following sections will outline the chronological path of inquiry that was guided by my subjective act of meaning making.

## 4. Getting to Know Oscar

I began writing about my teaching experiences more regularly during the COVID-19 pandemic. It was a chaotic year of teaching on and offline in a developing country with challenging political and physical difficulties. I intended to delve into my experiences with online learning and the impact online learning had on my relationships with students. After multiple explorations, I noted how my experiences with Oscar stood out.

*When I think of online learning during the COVID pandemic, I think of Oscar. I felt that being online helped me to get to know Oscar better. Beginning our relationship online gave me an intimate understanding of him that I don't believe I would have had in a normal classroom setting.*

I believed that teaching and learning were interdependent processes. This meant that I subscribed to the significant role that meaningful relationships played in education, on and offline. In Miller's (2021) [22] qualitative research on graduate-level teacher education during the pandemic, she found that online learning heightened in-service teachers' attitudes towards re(building) teacher–student relationships in schools. This suggested that online learning had compelled teachers to pay closer attention to teaching and learning relationships. Perhaps being online with Oscar compelled me to spend an increased amount of individual time with him, which is a luxury that few teachers have.

While unintended, schooling routines often reduce the time and opportunity for teachers to maintain close bonds with their students. With routines disrupted during online learning, it gave me time to develop ways to spend extra individual time with Oscar. Similarly, Fawns et al. (2019) [23] eloquently reported how strong educational relationships can be fostered regardless of physicality. This alluded to the connection that educators make with their students rather than attributing the quality of relationships to physicality. Relational labour takes time, effort and persistence and, in terms of Oscar's situation, being online gave me the chance to provide him with individual attention early in the school year.

> *Prior to starting online schooling, I had two early indicators that Oscar might require extra attention. One was from the school counsellor who described him as being a stubborn, smart but potentially difficult student. The other was from his father who emailed me to wish me luck getting to know his "active" son. These two alerts piqued my curiosity and led me to guess that he might need extra attention, not necessarily from an academic standpoint.*

I welcomed the school year with a strong desire to get to know my students as quickly as possible but, as Miller's (2021) [22] expressed above, the busyness of schooling makes it challenging to carve out time and opportunity to provide individual attention. Although previous impressions and/or (in)formal anecdotes from others may inform my view of students, I believed in developing my own understanding and impression of students. As an experienced educator, I was also acutely aware of the pivotal role that relationships play in learning. In Anderson and Taner's (2022) [24] metasummary of expert teachers, they described how expert teachers subscribed to the significance of "interpersonal relationships" and found ways to foster and demonstrate care and support. This metasummary also underscored the importance of relational labour in teaching because it underpins how teachers engage, motivate and manage learning behaviour. While I deemed myself to be an attentive teacher by being fluid, responsive and empathetic, I wondered if I would have adequate time to get to know him holistically.

> *Oscar was eight years old and he had been at this school for three years, working with the same group of students since Kindergarten. He was from a multilingual family who spoke primarily French and Italian at home. While he had an older sister, she was living elsewhere but visited from time to time. He was usually at home alone with his nanny and two cats. He was alone at home with his father, and removed from his extended family overseas. School was his means of socializing so COVID and being online exacerbated his feelings of loneliness.*

Delving further into Oscar's background allowed me to better appreciate the challenges that he was confronting in addition to online schooling. In a systematic review of the impact of lockdowns during the pandemic on child and adolescent mental health, Panchal et al. (2021) [25] indicated that increased levels of anxiety, loneliness and anger were found despite the increase in quality time within families. In Oscar's case, despite his father working from home at the time, he may not have been able to fulfill Oscar's need for similar-aged playmates. However, Oscar was from an expatriate family. Thus, feelings of loneliness and isolation in expatriate families are not uncommon; in fact, Sterle et al. (2018) [26] outlined the ways in which digital technology has ameliorated some of these feelings through virtual connectivity. Interestingly, at the time of the pandemic, Oscar and his father were novice users of social/digital media and the internet. Since stringent COVID-19 protocols at the time included lockdowns and quarantines, these had diminished Oscar's ability to arrange playdates or sleepovers at his house or elsewhere. These compounding factors, in addition to the new social, technical and learning protocols of online schooling, could have exacerbated Oscar's stress levels.

> *I still remember the first week of online schooling vividly. The first week was so chaotic that I wrote daily email updates to parents to smoothen out the process. Amidst the chaos, Oscar stood out because he was frequently disruptive throughout the day.*

My daily email updates were written to inform parents/families about my goals for learning progress and outcomes, amidst the chaotic interruptions within my online classroom. When Liao et al. (2021) [27] investigated the online practices of award-winning elementary teachers, they underscored the significance of parent and teacher relationships in successful online learning experiences. While the developmental needs of upper and lower elementary students varied, these teachers believed that parents or family members required clear roles and a good understanding of online learning expectations. These beliefs aligned with the guidelines provided by the U.S. Office of Educational Technology (2021) [28] regarding parental involvement; they stated that home support played a crucial role in supporting effective online schooling experiences. Similarly, research on Turkish parental experiences of pandemic-related online learning [29] highlighted that active parental involvement was needed to stand in for the academic, technical and physical support that teachers would normally provide at school. However, in adopting these teaching roles, parents felt that they carried the "burden" of teaching, which affected their relationships with their children adversely [29]. In Oscar's case, even though his father worked online in the next room during schooling hours, his father was unable to support him. Although he checked in with Oscar from time to time, Oscar demanded more attention than he was given. Even though his father made an effort to teach him digital literacy basics such as accessing Google Meet, the online platform used for schooling, Oscar was often unable to function independently online.

## 5. Oscar and His Father

Oscar's disruptive behaviour was one of many factors that contributed to the chaos that ensued in the first week of online school. While there was a larger mix of context-related challenges such as power and internet disruptions, slow internet connectivity, the lack of quiet and private spaces for online learning in some family settings, and the steep learning curve that sixteen eight and nine-year-old students with low digital literacy skills had to negotiate to learn, it was Oscar's first experience of being online.

> It was difficult to run lessons when he kept trying to talk above everyone. He did not want to work quietly even though he understood it was disturbing others. I would usually spend a lot of time during the first week of school with community-building activities that would help everyone to settle into a rhythm and routine of learning but persistent internet and power disruptions caused routine processes such as beginning lessons with a book or a pencil to become chaotic and disorganized because some students would drift in and out of connectivity or only have heard intermittent instructions. While Oscar's anger and frustration did not escalate, they did not dissipate either. Despite exhibiting negative emotions such as anger, frustration and impatience, Oscar was usually one of the first students who signed in punctually by 7:30 most mornings. I took this as one of few promising signs that he wanted to be connected to the group despite his continued struggles with online learning.

Needless to say, the first few weeks of online learning were challenging for most learners globally as face-to-face interactions were replaced by disembodied experiences. For younger students who tended to be less able to self-regulate during online learning [30], as was the case for Oscar's group, the need for explicit and clear routines was urgent. I believe that student-centred routines allow teachers to be less reliant on disciplinary or punitive approaches to managing learning behaviour [31]. In fact, I treated Oscar's disruptions as one of the many challenges that needed my attention and this helped me to be more tolerant of his outbursts. This acceptance helped the group to focus on community building and support rather than the multitude of issues that disrupted their learning. As reported in Yan et al.'s (2021) [30] age-delineated investigation of online learning experiences, younger students were more likely to gravitate towards synchronous interactions with adults such as teachers and parents, while older students tended to be more self or peer reliant. While Oscar's group demonstrated a largely positive desire

to remain connected to each other and me throughout the school day, Oscar's need for connection was slightly different from his peers.

> *I was constantly worried and dubious of the quality of Oscar's online learning experiences, especially when his video and microphone were turned off. When we first started online learning, he was not able to mute himself or turn off his camera but over time, he regularly muted himself and stopped his video feed. Some days, I wondered if he was online to be connected to us, or if he was forced to be online because his dad was working in the next room. Beneath this worry, I was concerned about his unmonitored access to the internet because I had very little insight into how he was spending his time during online learning.*

These ongoing uncertainties required me to employ Hochschild's (2012) [32] concept of deep acting, where a professional front of care, understanding and tolerance was projected because I believed that it was my responsibility to do so. After many years, deep acting becomes the adopted persona and the performer may lose sight of their authentic selves [32] (p. 37). While I was conscious of my worries, I was experienced enough to not act upon these emotions. Nyanjom and Naylor (2021) [33] described this as the "emotional suppression" [33] (p. 158) that teachers endure in the pursuit of professionalism. During the pandemic, online education redefined how educators performed emotional labour; thus, as teachers were learning how to teach online, they were also crafting their online teaching selves [33]. My conception of professionalism required deep acting that included emotional suppression that allowed me to prioritize my concerns for Oscar over my raw emotions. Over time, my suspicions about Oscar's inappropriate use of the internet were exposed.

> *During the six to eight weeks online, we got more comfortable with each other so I got to know Oscar a bit better, which resulted in him telling me about his favourite YouTube gamers. He described how he enjoyed watching gamers narrating and playing mature-rated games (i.e., for 18+ gamers). There was one occasion that we watched one of these videos together. During the video, I commented about the violence and gore in the video game. He laughed but was thoroughly fixated on the video and game. It was quite clear that the game and video were not meant for under 18 viewers as the video was peppered with expletives, rude gestures and sustained yelling.*

My initial reaction to Oscar's YouTube video habit was helplessness. I felt helpless because I did not think it was useful to report his behaviour to his father. Two thoughts guided my decisions at the time: I thought that it was irrational to expect teachers to "control" online behaviour and I assumed that Oscar's father would place the blame on me in terms of online behaviour management. My fear of negative repercussions stopped me from asking Oscar's father about his stance on online gaming and videos. Despite my knowing that parents needed to play a central role in managing online behaviour at home, I did not think it was realistic to expect Oscar's father to closely monitor Oscar's behaviour during online learning because Oscar's father also had to work online.

To understand the link between anxiety and online gaming behaviour of young children, De Pasquale et al. (2021) [34] studied Italian children aged 8 to 10 during the pandemic. They found that "parenting styles and family rules" [34] (p. 2) played a significant factor in reducing negative effects such as exposure to violence and aggression that could induce anxiety or antisocial behaviour. Interestingly, they also noticed how male children reported an increase in self-control and interaction amongst fellow gamers [34]. If Oscar's attraction to gaming videos is framed as a means to connect with others, it would allow for a more nuanced appreciation of how Oscar may have felt alienated from other children during the pandemic. At the time, with lockdown measures in place, Oscar was only able to interact physically with his father and nanny. By watching YouTube videos and playing online games, Oscar found a way to connect with others.

> *Since I did not want to break the fragile trust that we had built, I focused on dissuading and asking Oscar to think about the bad language he was listening to and getting excited about. When I asked if his father allowed him to watch these videos, judging from his downcast reaction, I think he knew that he did not have his father's permission. Even*

*though these moments were brief, they gave me a valuable insight into how Oscar was exploring the internet.*

At the time, I paid close attention to Oscar's behaviour because he frequently had outbursts of anger and frustration. Thus, when he exhibited excitement or any positive emotion, I felt compelled to investigate further into the trigger, context or situation for clues that could help me to better understand him. This heightened state could be interpreted as the anxiousness that I constantly associated with Oscar. Frenzel, Daniels and Burić [35] (p. 252) suggested that teacher anxiety was under-reported due to its negative connotation, and perhaps its synonymity to vulnerability and weakness. In Oscar's case, my anxiety stemmed from my inability to consistently engage him and influence his behaviour. However, in this encounter, his excitement was contagious. It was similar to what Frenzel, Daniels and Burić [35] (p. 252) described as "catching" feelings. Even though I did not agree with his online viewing preferences, I was caught up in the positive emotions that he displayed. At the time, I may have considered this a sign of trust, indicative of a developing bond between us.

The significance of physical proximity in managing behaviour of young children cannot be understated. For example, Oscar's downcast and perhaps guilty reaction to my question about his father's permission could be interpreted as the strong influence that his father had on him. Additionally, even though I was online or virtually present with Oscar, my influence on his online behaviour was negligible in comparison to his father's. Rawlings et al. [31] (p. 407) described proximity as a means to communicate and appease students physically. When working with younger students, such as Oscar, physical proximity can be reassuring as much as it can be intimidating. My inability to influence Oscar's behaviour online supported Liu et al.'s (2010) [36] assertion that physical proximity in online learning for younger learners was a necessity. In their study of pandemic online learning, Lau and Lee (2021) [37] also emphasized the key roles that parents or families played in order to guide online learning behaviour.

Without adequate parental support for young learners online, unfettered access to the internet could have dire consequences. By being visible or readily accessible, parent proximity would mimic the level of close supervision that younger students would generally experience in school. While I sometimes reminded Oscar of his father's physical proximity, this reminder was used sparingly. I was cautious with this approach and applied it when he was extremely uncooperative. In the early weeks of online learning, I might have also assumed that Oscar's father's proximity was an adequate substitute for my presence, but I quickly learnt otherwise. Chen (2021) [38] stated that teacher emotions can impact their efficacy. I felt a loss of efficacy due to the continued sense of helplessness that dominated my relationship with Oscar during online learning. When we returned to face-to-face learning, it was quickly evident to me how much physical proximity made a difference for Oscar. There were far more instances where I could deter his YouTube and gaming habits in class by standing next to him or engaging him in other actions or conversations. In person, I felt that there was a higher chance that I was more able to influence Oscar's online behaviour.

*I believed in empowering rather than enforcing so my gradual and persuasive approach of teaching Oscar to self-manage his use of the internet yielded slow and inconsistent results. Most days Oscar could be persuaded to work independently on tasks but on rougher days, when he was tired or irritable, there was little that I could do to engage him. On those days, I felt guilty because I felt that I had let Oscar and his dad down. According to Oscar, he felt that his dad wanted me to be stricter and to be less indulgent of him. At the time, I felt that there was little that I could do on days that Oscar hung up, turned off his video or when he was muted but online in class. In hindsight, I wondered if I could have done more, perhaps I could have emailed his father each time any of these behaviours occurred. Even though this guilt tapered over time, it still eats away at me and I often still wonder if I had made the right choices with him.*

Anxiety, insecurity and inadequacy permeated my thoughts of online learning with Oscar. For some teachers, being online reminded them of the importance of teacher–student relationships [22] and I can rationalise that being online expedited the process of getting to know Oscar individually. As a teacher, I often actively sought varied methods to make meaningful and healthy connections with my students because I believed that teaching is a relational act. In a longitudinal study of high-school student engagement, Martin and Collie [39] (p. 872) investigated the cumulative influence of teacher–student relationships to show how these bonds impact student learning behaviour. They recommended the cultivation of positive relationships rather than attempting to mitigate the potential implications of negative relationships [39] (p. 861). Since I normally focused on building positive, caring and honest relationships with my students, Oscar's outbursts and unpredictable behaviour made it challenging to nurture consistent positive interactions. If the disembodied nature of online learning can increase students' need for attention and support [40], there was also the possibility that Oscar's behaviour was a means to seek attention from his teacher and peers.

I consciously made an effort to maintain positive interactions with my students because I believed in the importance of decreasing the probability of Dewey's [41] (p. 8) "miseducative" or inhibiting learning experiences for Oscar. It is not novel to state that teachers play a crucial role in the lives of their students; their emotions can affect and have an effect on their students [38]. Since online learning achievement and satisfaction are closely related to the quality of relationships [42], I wondered if my inability to manage Oscar and his behaviour online produced "miseducative" experiences that were harmful rather than harmless.

These purposeful choices and musings shaped my emotional management of inward and outward displays of emotions. Burić and Frenzel (2021) [43] theorized that teachers perform emotional labour within invisible "emotional labor rules" to ensure that the expected behaviour is exhibited. In this case, I continued to project an optimistic front that urged and cajoled Oscar to behave appropriately despite having little faith that these actions were influencing his behaviour and choices. This optimistic front was informed by the impression that I had of teachers being relentlessly hopeful, positive and encouraging.

When teacher behaviour is framed as emotional labour, it makes the rules that teachers may unquestioningly follow to regulate their actions, emotions and expressions visibly. Burić and Frenzel (2021) [43] stated that teachers hid their feelings and faked emotions for different purposes. I utilized both strategies in my relationship with Oscar. For example, I had to feign excitement and interest when he shared YouTube gaming videos that contained images, language and actions that I would not normally view. These fake emotions were necessary to gain his trust. My true emotions would have been surprise, shock and horror at the violence that this eight year old was viewing. Instead, I chose to suppress these feelings in order to gain an insider view of his online behaviour.

Over a prolonged time, these emotional management strategies that employed surface acting could have adverse effects on teachers in the long run because they would be cognitively and physically demanding [43]. I cannot deny that my relationship with Oscar was mentally draining because I continually questioned my efficacy, choices and actions. For example, I often wondered if Oscar's exposure to the internet would have been delayed if he did not need to be online unsupervised at eight years old. I also questioned if there were worthwhile online learning moments that could counter the cost of exposing Oscar to the inappropriate content, language and behaviour he had seen and heard. I was acquiescing to inappropriate behaviour to prioritize my relationship with Oscar. The dissonance between my behaviour and beliefs was jarring, but at the time, it was a choice that I made in order to understand Oscar. It was more important for me to gain a deeper insight into his behaviour than to reprimand, which could have caused irreparable damage to our relationship. These gnawing doubts did not disappear when we returned to normal, face-to-face learning because Oscar continued to use bad language and had angry outbursts on the playground and school bus.

Over time, Oscar's father voiced his concern over his escalating behaviour and argued that these were school-acquired bad habits. Although Oscar's father's complaint was presented with a positive and problem-solving intention, it was clear that he blamed me, the teacher, for being unable to control his son's behaviour and choice of words. When I spoke to Oscar's father, I employed deep acting strategies that allowed me to project confidence and empathy. Based on previous teacher–parent encounters, I learned not to show vulnerability because it equated to culpability and weakness. My experiences with other parents also taught me that some parents needed the teacher to take on the blame in order to find a solution for their child(ren). My rational self could rationalise that Oscar's behaviour was not entirely my fault, but I could not avoid feeling hurt and helpless. In order to move forward, I have learned to accept that sometimes it was unavoidable to be viewed negatively by parents. To move beyond the stress of uncertainties, challenges or setbacks, I hung onto my belief in the strength of meaningful learning relationships with my students and parents. Making a conscious effort to ground myself in my beliefs and values of education helped me to reflect, heal and move forward.

Through this process of writing to reflect, analyse and understand my relationship with Oscar, I made time and space to flush out my raw and repressed emotions about him. Tangled within these negative emotions were also moments of joyful, positive and happy moments where Oscar and I joked, giggled and bonded over mutual interests such as anime characters.

## 6. Making Sense of My Relationship with Oscar and His Father

I went through a few long pauses when I chronologically explored the moments I had with Oscar. Reliving those memories stirred up the unhappy parent emails, lengthy seven-and-a-half-hour days online with 18 eight and nine-year-old students, and the challenges of living in a developing country with consistent power and internet failures. My mind projected vivid images of me sitting in my living room staring at my computer screen and my body remembered the kinks in my neck and the stretches that I did during breaks. It was still early in my honeymoon phase of resettling in a new country, at a new school and attuning myself to the group of eager third-grade students.

Nevertheless, I could still smile when I pictured Oscar's face, with his unruly hair, large grin and booming voice over the computer. It was easy for me to develop a soft spot for him from day one of school. His need for attention was palpable: from acting out to disrupt the group, which elicited negative attention from his peers, to cracking jokes (appropriate and inappropriate) as a way to fit in. Interestingly, when I dived into these memories, I noticed how I felt a continued sense of guilt that I had not done enough to help him.

Teasing out these simmering emotions and thoughts involved naming, describing and conjecturing. Throughout this analysis process, I felt a need to recognise and identify my fleeting thoughts and buried emotions. Identification allowed me to seek and play with words to try to describe the thoughts and feelings that enveloped the memory. By assigning words to my embodied experience, I "tasted" words and allowed them to "percolate" and "ripen" in order to evaluate how they represented how I felt and thought.

This analysis process was guided by curiosity and Hochschild's (2012) [32] work on feeling rules. Separating emotions and feelings entailed defining emotions as the visceral experience or the mental state of feelings, and feelings as the visible or performance of emotions. The rawness or unfettered emotions were difficult to ascertain and recall because they had been numbed over time. Additionally, tuning into the feeling rules or management process that I had routinized as a teacher necessitated pauses, distancing and finding the frequency of "me" within the experience.

*I was patient, was that enough?*

*I was curious, was curiosity helpful or should I have been firmer?*

*I focused on him, but did my student-centred approach exacerbate his need for attention?*

*I tried to include him, but was he learning anything from me?*

The naming of emotions and the ensuing feelings was generative and exhausting due to the examples of dichotomies displayed above. However, these dualities helped me to explore the conceptualisations within Hochschild's (2012) [32] "surface and deep acting". I wondered about the days when I needed to taper my emotions and tap into my impression management skills. My belief that teachers needed to be caring and patient fed into my need to repress any emotional turmoil I had in order to project the image of a calm, persevering and encouraging teacher. When such efforts into surface acting are performed habitually, they may blur into deep acting [32] over time or result in mental breakdowns. My internalized self-talk in both forms of acting sought to align the emotions that surfaced with my rationalised feelings. Through this process of analysis, I discovered that my impression management efforts to convince myself that it was imperative to manage my emotions (internal) and feelings (external) to uphold my idealized image of a good teacher had inadvertently disconnected me from most of my authentic emotions. It made me wonder if it was realistic for teachers to be authentic with their emotions and feelings.

It is undeniable that the socialized nature of teaching requires teachers to be adept at managing their emotions and resulting feelings. In their exploration of emotional labour and teacher burnout, Bodenheimer and Shuster (2020) [44] theorized about the likelihood of mental exhaustion when teachers over utilize Hochschild's (2012) [32] surface and deep acting as a means to cope with the pressures of teaching realities. Surface acting can be imagined as a way to conceal or conjure emotions in order to deceive others. On the other hand, we employ deep acting as a way to rationalise and justify the expected emotion by deceiving or denying ourselves our raw emotions. These emotional constructs illuminate the emotional labour teachers engage in daily. By suggesting "emotional autonomy", Bodenheimer and Shuster [44] (p. 73) alluded to the notion that the surface and deep-acting mechanisms that teachers apply may be shaped by external forces such as pressures from administrative measures or educational policies. When emotional autonomy is promoted, it can empower teachers to question internal and external pressures and galvanize existing discourse on more humanistic ways to decipher teaching lives. Such discourse would inspire teachers such as me to be more authentic with emotion and feeling work.

Learnings from this reflective and reflexive process helped me to differentiate between the lingering emotions that I had about Oscar and his father. This critical understanding prompted me to reconsider the emotions and feelings that were woven into the fabric of my stories about Oscar. Helping teachers tease out the differences between emotions (invisible) and feelings (visible) could be useful when learning how to manage or negotiate difficult relationships with students and parents. Recognising the differences between inner thoughts and outer actions could help teachers get to know themselves better, especially when they are under duress or managing stressful situations.

I am uncertain if being authentic with one's emotions and feelings would be the ideal equilibrium for teachers given their powerlessness to change or control larger, external influences on their lives. These influences may require teachers to repress their emotions and feelings, producing acquiescence that materialises as surface or deep acting; however, if these are unaddressed, teachers may wind up feeling burnt out or exhausted [45]. In their study about emotional labour and burnout, Näring, Briët, and Brouwers [45] mentioned that a way forward for emotional labour research could be to explore aspects of education that enable teachers to thrive and be empowered. Identifying supportive factors that could taper the negative effects of prolonged surface acting, which has been shown to increase depersonalization and dissonance in teachers [46] will be beneficial.

Likewise, encouraging teachers to learn how to navigate and manage their emotions as an integral part of their professional development could also equip teachers for the increasingly demanding responsibilities that teachers confront daily [46]. Teaching requires mental fortitude because teachers are required to manage their feelings and emotions, as well as teach others to manage theirs [47]. Online teaching during COVID-19 heightened our need for physical connection and it also highlighted the assumed but invisible role that

teachers play in cultivating healthy relationships and emotions at schools. What is clear from this study is the value of promoting professional development opportunities that can help teachers to understand and acquire skills that may help them to manage the impact of their emotions and relational labour. Purposeful, reflective and reflexive examinations of experiences may help teachers to get to know themselves better and offer timely space for teachers to work through their thoughts and memories.

Identifying supportive factors that could taper the negative effects of prolonged surface acting, which has been shown to increase depersonalization and dissonance in teachers [46], will be beneficial. Encouraging teachers to learn how to manage their emotions as an integral part of their professional development could also better prepare them for the increasingly demanding responsibilities that teachers confront daily [46]. What is clear from this study is the need to encourage teachers to reflect on their experiences so that they may better understand and manage the potential emotional strain that they may have experienced. Purposeful reflective and reflexive examinations of experiences may help teachers get to know themselves better and offer a timely space for them to work through their thoughts and memories.

### 7. Closing Reflections

This writing contained an exploration of one teacher's storied experience of online learning during the COVID-19 pandemic. It highlighted the possibilities that teachers can consider when they attempt to make sense of their relationships and responsibilities to students and parents. The intentional weaving of these particularities produced a nuanced, authentic and contextualised narrative of how online learning experiences can impact the lives of students, their families and teachers within and beyond the classroom.

Since the aim of this study was to illuminate the meaning-making process that teachers can employ to understand their experiences, it is pertinent to state that the conjectural insights presented were designed to promote dialogue amongst teachers, teacher educators, educational researchers and policymakers who seek to ameliorate the complexities of educational relationships in online education. Through the idiosyncratic narrative lens on teaching lives, this paper supplements existing narrative work on the subtleties and challenges within teaching lives.

**Funding:** This research received no external funding.

**Institutional Review Board Statement:** Not applicable.

**Informed Consent Statement:** Not applicable.

**Data Availability Statement:** Data are contained within the article.

**Conflicts of Interest:** The author declares no conflict of interest.

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
