# Peer review of "A Narrative Exploration of One Teacher’s Storied Experiences of Online Learning during COVID-19"

_education, doi:10.3390/educsci13121211_

Round 1

Reviewer 1 Report

Comments and Suggestions for Authors

Bravo to the author for this interesting and reflective study! I thought this manuscript was well organized and articulated with excellent references to relevant literature. Through the telling and retelling of storied experiences while teaching online, the author invites educational researchers, teachers, and other stakeholders to engage in deeper conversations around the possibilities and importance of reflective practices as we think about online teacher-student-family relationships. The excerpts of field notes sliding temporarily backwards and forward in time were compelling as the author invited us into her world of online teaching while coming alongside Oscar and in many ways, his father. While the study does not aim to provide answers or how to’s, it was as the author suggested,  “underpinned by the notion that teachers can write, reflect and  muse in order to make sense of challenging experiences.” It is in these intentional inquiry spaces that we can begin to relive and re-imagine more relational ways of coming alongside students, especially increasingly digital and online spaces.

The author noted that this study was an autoethnography. I did wonder about this as I understood this study to be more of an autobiographical narrative inquiry rather than an autoethnography. The latter being more focused on understanding a wider cultural, political, and social phenomenon. I encourage the author to read more about the distinctions between an autobiographical narrative inquiry as shared by Clandinin and Connelly (2000) and autoethnography; there are similarities but also different nuances between the two.

Overall, a provocative manuscript that is timely and thought-provoking. There were no major revisions required that I could see. A minor suggestion is to use semicolons consistenly for the in-text citations where multiple authors are referenced and listed.

Author Response

Thank you for your kind words of encouragement.

I have corrected the semi-colon errors as indicated. 

I have also clarified my methodology section - re autoethnography. 

I look forward to your feedback. 

Reviewer 2 Report

Comments and Suggestions for Authors

The article appears very interesting and substantive. The autoetnographic perspective on narrative inquiry deepening the knowledge of the Self has emotional - analitic, and to some extent, the cooperative dimension grounded in an appropriately described setting and context.  Properly conducted process of reflection enhances the quality of the paper and fosters reflective thinking about building the valuable relationships between the Author, Oscar and his parent, what requires the mature person-centered reference to it.  The article also elicits the meaning of teacher assistance for students who can get lost in the world that not necassary supports them. What can be improved a bit in the article is to: build clear, concrete particular conclusions at the end of empirical narrative analysis;  form more elaborated  final conclusions and implications; refer more clearly to the narrative components. Also, it might be a good idea to refer to Michael Bamberg's concept on narrative analysis.  

Author Response

Thank you for your words of encouragement. 

I have clarified my methodological and closing/concluding sections. 

I look forward to your feedback. 

Reviewer 3 Report

Comments and Suggestions for Authors

The paper reflects on a teacher’s experience of relationships between herself, a student and parent in the context of online teaching during COVID lockdowns. The paper draws on relevant literature and has merit. It has been written in a fluid style that facilitates reading.

The Introduction and Methodology sections discuss narrative inquiry in broad terms but more precision in the use of terminology (story-telling, storying, story not the same as narrative, retelling my story, narrated experience, restorying) is required. Similarly, the claim that an “autoethnographic lens” (L115) is applied to this narrative study seems superficial without any reference to the expectations of autoethnography as a lens, or the procedural stages of autoethnography as a research method. The challenge here is that the methodological procedures need to be more convincingly presented so that the reader has confidence in the soundness of the research and its value.

The presentation of the research needs to more convincingly embody the unfolding of the narrative inquiry and the development of ideas. Where do the italicised sections of text fit in the inquiry process? When were they written? As it is, they almost appear to be written for the purposes of the paper.

I am not sure that the real issue emerging from the inquiry has been identified. I would have thought the ethical dilemma experienced by the teacher regarding the child watching inappropriate videos is the significant matter that has emerged from this inquiry. This needs to be interrogated at greater depth.

L137 –The circumstances of online teaching during COVID lockdowns is diverse so more information about the context is needed here (as well as appearing later when woven into the narrative). How old were the children? How long had the teacher known them? How was the online option being used as a substitute for regular schooling?

L470 faint should be feign.

Comments on the Quality of English Language

Writing style and English language usage is sound.

Author Response

Thank you for providing your sound critique, and for catching me “feign” rather than faint in embarrassment.  

I have rewritten several sections to address your points: introduction, methodology and conclusion. 

I look forward to your feedback. 

Round 2

Reviewer 3 Report

Comments and Suggestions for Authors

Line 547 - I think you mean "temper' rather than "taper"